# Overcoming Cancer Resistance: Strategies and Modalities for Effective Treatment

**DOI:** 10.3390/biomedicines12081801

**Published:** 2024-08-08

**Authors:** Mahesh Koirala, Mario DiPaola

**Affiliations:** Therabene Inc., Norwood, MA 02062, USA; mahesh@therabene.com

**Keywords:** cancer drug resistance, targeted protein degradation, immunotherapy combinations, precision medicine, novel drug delivery systems, PROTACs, SNIPERs, combination therapies

## Abstract

Resistance to cancer drugs is a complex phenomenon that poses a significant challenge in the treatment of various malignancies. This review comprehensively explores cancer resistance mechanisms and discusses emerging strategies and modalities to overcome this obstacle. Many factors contribute to cancer resistance, including genetic mutations, activation of alternative signaling pathways, and alterations in the tumor microenvironment. Innovative approaches, such as targeted protein degradation, immunotherapy combinations, precision medicine, and novel drug delivery systems, hold promise for improving treatment outcomes. Understanding the intricacies of cancer resistance and leveraging innovative modalities are essential for advancing cancer therapy.

## 1. Introduction

Cancer represents a major healthcare crisis; globally, in 2020, there were approximately 18 million cases of cancer, leading to 10 million deaths [1]. It is estimated that in the next decade, the incidence of cancer will rise to more than 27 million, with cancer-related deaths exceeding 16 million [2]. A major problem in combatting cancer is drug resistance, which remains a formidable challenge in oncology, limiting the effectiveness of conventional and targeted therapies [3]. Despite advances in treatment modalities, resistance mechanisms continue to undermine therapeutic efficacy, leading to disease recurrence and poor patient outcomes [4]. In this review, we outline the various identified modes of drug resistance, which include drug efflux, drug target alterations, DNA repair, cell death inhibition, evasion of drug action, microenvironmental changes, and an epithelial-to-mesenchymal transition (Figure 1) [5]. The articles considered in this review span research from the recent years, ensuring a comprehensive and up-to-date analysis of the current state of cancer resistance and the emerging strategies to combat it [6].

### 1.1. Mechanisms of Cancer Resistance

#### 1.1.1. Drug Efflux

Drug efflux is a major mechanism by which cancer cells develop resistance to chemotherapy and other therapeutic agents. This process involves the active transport of drugs out of the cancer cells, leading to a reduction in the intracellular concentration of cancer drugs to sub-lethal levels, resulting in ineffective treatment.

Drug efflux is primarily driven by a family of proteins known as ATP-binding cassette (ABC) transporters.

ABC transporters use the energy from ATP hydrolysis to actively transport various molecules, including cancer drugs, across cell membranes against their concentration gradients. The expression and activity of ABC transporters are typically high in certain tissues as a mechanism of protection from toxins. However, in cancer cells, the overexpression or increased activity of these transporters can lead to significant challenges in treatment, as it effectively removes therapeutic agents from the cells before they can exert their cytotoxic effects.

ABCB1, ABCC1, and ABCG2 transporters are the main contributors to multidrug resistance (MDR) in cancer chemotherapy [7]. In fact, several studies have shown that overexpression of MDR1 has been implicated in drug resistance for different cancers, including colorectal, lung, breast, and prostate, among others [8,9,10,11].

#### 1.1.2. Drug Target Alterations

Drug target alterations represent a critical mechanism through which cancer cells develop resistance to targeted therapies. Drug target alterations may result from secondary mutation(s) in the target protein to epigenetic alterations, yielding elevated protein expression. Targeted therapies work by specifically inhibiting proteins that are essential for the growth and survival of cancer cells. However, when these target proteins mutate or undergo structural or functional changes, the effectiveness of the therapy can be significantly diminished. 

A case in point for this type of resistance is the epidermal growth factor (EGF) receptor and its inhibition through the use of kinase inhibitors as treatment for NSCLC [12,13]. While there was a strong initial response to EGF receptor inhibition, as many as 50% of the responding patients developed resistance to first- and second-generation inhibitors within one year [14,15] because of specific receptor mutations affecting the binding of inhibitors. Redesigned third-generation inhibitors were able to overcome the inhibition [16,17], but, again, after some use, resistance was reestablished through a similar mutation mechanism [18], necessitating a fourth-generation inhibitor [19,20].

Similarly, the use of tamoxifen in the treatment of breast cancer has led to significant resistance. The mechanism of resistance is thought to be a result of the overexpression of RTKs (receptor tyrosine kinases) and the activation of the PI3K-PTEN/AKT/mTOR [21]. Additionally, the truncated isoform of ERα66, known as ERα36, which is located on the cytoplasmic membrane of breast cancer cells [22,23], has been linked to drug resistance and metastasis in breast cancer [23,24]. Tamoxifen is thought to activate ERα36, which then triggers the MAPK, AKT, and other signaling pathways, leading to tamoxifen resistance [25]. To bypass such resistance, aromatase inhibitors, inhibiting the last step in estrogen synthesis, were developed. This new class of molecules is currently used as primary therapy in breast cancer [26].

#### 1.1.3. DNA Repair

Older chemotherapy drugs, such as oxaliplatin/cisplatin or 5-fluorouracil (5-FU), induce DNA damage as a mechanism leading to cancer cell death. Unfortunately, the cellular DNA damage response (DDR) process will repair DNA lesions of affected cells to the anti-cancer drugs, thus resulting in reduced efficacy of the drugs and drug resistance [27]. Genes involved in DNA repair, such as FEN1, FANCG, and RAD23B, have been found to be upregulated in 5-FU-resistant human colon cancer cell lines [28,29]. Furthermore, 5-FU treatment has been shown to upregulate p53-target genes involved in the DNA damage response and repair, leading to reduced cell cycle arrest and apoptosis in resistant cell lines compared to parental cell lines [29]. Although disrupting DDR might reduce resistance induced by DNA repair, it could also increase the risk of new mutations due to genomic instability, potentially initiating new rounds of carcinogenesis. Therefore, the DNA damage response is a complex mechanism in cancer treatment and recurrence that requires careful consideration when used as an anti-cancer strategy. 

In addition to gene mutations that activate DNA repair genes, epigenetic regulations such as DNA methylation, histone modifications, and miRNAs can also affect the expression of DNA repair genes, contributing to drug resistance. However, it is important to note that there is a complex interplay among DNA repair, DNA methylation, and cancer resistance. Targeting DNA methylation might offer a promising approach to overcoming drug resistance in cancer.

#### 1.1.4. Cell Death Inhibition

Cancer cells are known to regulate apoptotic pathways tightly. Apoptosis not only removes infected cells from the body but also supports the immune system and maintains homeostasis [30]. Caspase proteolytic enzymes play a central role in the mediation of programmed cell death. It has been widely reported that caspase mutations during tumor therapy have resulted in chemoresistance. Specifically, caspase-8 mutations have been detected in gastric cancers. In these cancers, the procaspase-8Q482H mutation disrupts the dimerization of procaspase-8 protein monomers, thus preventing apoptosis and leading to chemotherapy resistance [31].

Another mechanism for cell death inhibition observed in many cancers is through the overexpression of anti-apoptotic BCL2 family proteins. Increased expression of BCL2 results from the t(14;18) chromosomal translocation in follicular lymphoma [32] and diffuse large B-cell lymphoma [33]. While this translocation is rare in solid tumors, BCL2 protein overexpression has been observed in some breast and prostate cancers [34,35,36]. Other mechanisms of BCL2 overexpression include transcriptional activation by NF-κB signaling [37] and promoter hypomethylation [38].

Several studies have shown that MCL1 and BCL2L1 (BCLXL) are frequently overexpressed in various tumor types [39,40]. Increased MCL1 transcription can result from DEK transcription factor amplification [41] or constitutive activation of STAT3 [42]. Additionally, post-translational mechanisms may also result in the elevated expression of antiapoptotic BCL2 family proteins, such as MCL1 protein overexpression due to genetic inactivation of the ubiquitin ligase complex protein FBW7 [43].

DNA damage typically activates the tumor-suppressor p53, leading to apoptosis through the upregulation of proapoptotic genes, like PUMA, NOXA, BID, and BAX [13,44,45,46]. However, TP53 is one of the most altered genes across all cancers, and loss of TP53 leads to the potentiation of tumorigenesis in multiple murine cancer models [47]. PUMA is a key mediator of p53-induced apoptosis in response to DNA damage; therefore, loss of p53-induced expression of BH3-only proteins like PUMA may contribute to disease progression. For example, decreased PUMA expression has been observed in melanoma [17]; in another study, BRAF-mutant melanomas were found to have impaired expression of p53 target genes, indicating a direct link between loss of p53 signaling, downregulation of PUMA, and melanoma disease progression. 

External pathways through membrane receptor signaling can also drive cell death inhibition, unlike the internal pathways discussed above. The expression of proapoptotic BH3-only proteins is regulated by external growth-promoting signaling pathways. Oncogenic kinase hyperactivation can lead to reduced expression of BH3-only proteins or function by suppressing transcription or via post-translational modifications that decrease protein stability or cause sequestration away from mitochondria. For instance, ERK phosphorylation of BIM leads to RSK1/2-sensitive, βTRCP-mediated proteasomal degradation [19,20], suggesting that hyperactivation of MAP kinase signaling may enable cancer cells to suppress BIM levels and evade apoptosis. Similarly, BAD phosphorylation by AKT and MAPK promotes binding to 14-3-3 proteins and sequestration [26,27,29]. PUMA expression can also be regulated by growth factor stimulation via PI3K and FOXO3A [48]. Thus, suppressing BH3-only protein activity through MEK–ERK and PI3K–AKT signaling pathways may be crucial for cancers driven by constitutively activated oncogenic kinases, such as EGFR [49,50,51], BRAF [52], KRAS [53], and BCR–ABL [54]. 

#### 1.1.5. Evasion of Drug Action

Cancer cells can alter their metabolic processes to support rapid growth and survival. Normal cells generate energy through oxidative phosphorylation (OXPHOS) in mitochondria, whereas cancer cells rely on aerobic glycolysis, even in oxygen-rich environments. This phenomenon displayed by cancer cells is referred to as the Warburg effect [55]. The Warburg effect promotes cancer cells’ growth and survival, facilitates adaptation to tissue oxygen concentration changes, and uses glycolysis intermediates to sustain cell proliferation. Additionally, glycolysis-produced lactate maintains the microenvironment in an oxidized state, aiding cancer infiltration and immune evasion [56]. Recent studies have shown that OXPHOS and mitochondrial metabolic processes are also crucial for cancer metabolism. In some tumors, OXPHOS increases through glucose, protein, amino acids (e.g., glutamine or tryptophan), and fatty acids oxidation. Other tumors utilize waste products like ammonia and lactate as energy sources [57,58,59,60,61,62]. Changes in cancer metabolism are controlled by carcinogenic mutations of key proteins, including MYC, PTEN, AKT, and PI3K. Additionally, carcinogenic mutations are regulated by changes in surrounding factors or specific metabolic enzymes like IDH1, SDH, and IDO [63,64,65]. 

#### 1.1.6. Microenvironmental Changes

Tumors do not comprise homogeneous types of cancer cells but are made up of various types of cells and an extracellular matrix (ECM) that work synergistically to enhance the survival and growth of cancer [65]. Indeed, the microenvironment of solid tumors includes ECM, fibroblasts, immune and inflammatory cells, blood vessels, and various nutrients and signaling molecules. 

The tumor microenvironment (TME) can contribute significantly to the intrinsic resistance against anti-cancer therapies. One critical factor in the TME is pH. Normally, extracellular pH (pHe) is slightly higher than intracellular pH (pHi), with values of approximately 7.3–7.5 for pHe and 6.8–7.2 for pHi [66]. However, cancer cells tend to develop a “reversed pH gradient,” characterized by elevated intracellular pH and reduced extracellular pH. This occurs through the action of proton pumps and the modulation of pH sensors [67,68]. The acidic extracellular environment (pH 6.5–7.1) in cancer cells is a known contributor to chemotherapeutic resistance [69]. This reversed pH gradient impairs the distribution of weak base anti-cancer drugs, a phenomenon known as “ion trapping,” which helps cancer cells evade apoptosis [70]. Leveraging this characteristic, therapies targeting the acidic microenvironment, such as proton pump inhibitors (PPIs), have shown efficacy in tumor reduction and the sensitization of cancer cells to chemotherapy. For instance, lansoprazole, a PPI, has demonstrated synergistic effects when combined with paclitaxel in melanoma cells, both in vitro and in vivo [71].

Post-treatment changes in the TME also play a role in cancer cell adaptation to chemotherapy or targeted therapies, thereby reducing drug efficacy and inducing resistance. For example, tumor-associated macrophages (TAMs) play a role in the acquisition of resistance in response to anti-cancer therapies in glioblastoma multiforme (GBM), a severe type of brain tumor [72]. Macrophages secrete high levels of colony-stimulating factor-1 (CSF-1) in GBM tumors, promoting cancer cell proliferation and survival. Consequently, the CSF-1 receptor (CSF-1R) has been targeted by small-molecule inhibitors or antibodies in cancer treatments, showing promising effects in vivo [73,74]. However, over 50% of GBM patients experience recurrence, which is driven by increased secretion of insulin-like growth factor-1 (IGF-1) from TAMs and IGF-1-induced activation of the phosphatidylinositol 3-kinase (PI3K) pathway in GBM cells [72]. Combining CSF-1R inhibition with IGF-1 receptor or PI3K pathway inhibition has been shown to extend overall survival in mouse models [72]. Thus, combined therapies targeting both cancer cells and the TME may significantly enhance anti-cancer efficacy by reducing drug resistance.

Another pathway for drug evasion is through enzymatic inactivation of the cancer drugs. For example, increased expression of glutathione S-transferase in cancer cells can conjugate drugs with glutathione, making them less effective. Several chemo agents, typically used as induction therapies/remission, such as cisplatin, carboplatin, cyclophosphamide, and doxorubicin, have been found to be adversely impacted by conjugation with glutathione [75].

It is clear that the tumor microenvironment can have a significant effect on tumor progression and therapeutic resistance; therefore, a better understanding of the TME and its interaction with tumor cells could substantially enhance therapy response and achieve better clinical outcomes with appropriate manipulation and targeting of the microenvironment.

#### 1.1.7. Epithelial-to-Mesenchymal Transition

The epithelial-to-mesenchymal transition (EMT) is a process that leads to tumor cell detachment from epithelial tissue, followed by dissemination and metastasis [76]. While EMT drives the initiation of metastasis in carcinomas, it is not clear whether EMT plays a similar role in sarcomas. However, there is increasing evidence that EMT does play a critical role in chemotherapy resistance. Recently, Debaugnies et al. [77] reported that the small Rho GTPase, RHOJ, is a key regulator in promoting resistance to a broad range of chemotherapeutic compounds in EMT tumor cells. They found that RHOJ regulates drug resistance associated with EMT by a response enhancement to replicative stress, along with the activation of DNA damage response, thus enabling the tumor cells to quickly repair any DNA damage caused by the chemotherapeutic agents. 

Furthermore, it has also been shown that EMT is driven by several key signaling pathways, including Notch, Wnt, transforming growth factor beta (TGFβ) and Hedgehog [78]. For example, the TGF-β signaling pathways have been correlated with the gain of drug resistance [79,80]. In fact, inhibition of TGF-β can reverse the process of EMT and increase the sensitivity of cancer cells to chemotherapies [79,80]. Wnt/β-catenin has been shown to be upregulated in human cancers, causing an acceleration in EMT-mediated metastasis and drug resistance [81]. Mechanistically, the stimulation of the Wnt signaling pathway by the binding of Wnt ligands into Frizzled receptors results in the enhancement of β-catenin accumulation in the cytoplasm, followed by translocation into the nucleus and potential activation of EMT. The Wnt/β-catenin/EMT axis has been implicated in augmenting the metastasis of both solid and hematological tumors, promoting the malignant behavior of tumor cells and resulting in therapy resistance. There are several studies in which Hh signaling has been implicated in cancer stemness or a mesenchymal phenotype. More specifically, the process of Gli1 induction of Snail, along with nuclear localization of β-catenin and subsequent EMT, has been reported in several cancer types, such as ovarian, breast, and basal cell carcinoma [82]. Canonical Hedgehog (Hh) signaling is known to enhance the cancer stem-like phenotype in pancreatic cancer, as Smo knockdown (smoothened) results in a decreased epithelial–mesenchymal transition (EMT), self-renewal, and resistance to gemcitabine [83]. Non-canonical Hh signaling activation also contributes to EMT through TGF-β signaling in the mesenchymal state [84]. Notably, there seems to be a positive feedback loop between EMT and Hh activation, where the loss of E-cadherin leads to increased expression of Shh (Sonic Hedgehog), Ptch (Patched), Smo, Gli1, and Gli2.

EMT-inducing transcription factors (EMT-TFs) are crucial in promoting drug resistance. In fact, the overexpression of EMT-TFs, such as Twist, Snail, Slug, ZEB, and FOXC2, has been associated with induced drug resistance [85,86,87,88]. Recent research has demonstrated that knocking out EMT-TFs Twist1 or Snail1 enhances sensitivity to gemcitabine and improves the survival rate in pancreatic ductal adenocarcinoma-bearing mice treated with the drug [89]. Some EMT-TFs promote resistance by increasing drug efflux via ABC transporters; this is supported by data showing EMT-TF binding sites on promoters of genes encoding ABC transporters [90]. Additionally, the overexpression of Twist, ZEB1/2, Slug, and Snail increases the expression and activity of ABCB1, contributing to drug resistance [90,91,92]. ABCG2, another ABC transporter closely linked to multidrug resistance (MDR), is regulated by Snail, MSX2, SOX2, and ZEB1 [93,94,95,96]. Other ABC transporters involved in MDR, such as ABCC1 and ABCC2, along with ABCC4 and ABCC5, have also been demonstrated to be modulated by EMT-TFs [97,98,99]. In fact, depletion of either FOXM1 or ABCC5 resulted in a decrease in drug efflux and a concomitant increase in cell death from paclitaxel [100] treatment. 

In addition to EMT-TFs, miRNAs, 20–24-nucleotide structures, are believed to be involved in linking EMT and ABC transporters [101]. MiRNAs regulate ABC transporters through disparate mechanisms, with some miRNAs acting at post-transcriptional levels by binding at the three prime untranslated regions (3′-UTR) and other miRNAs modulating transcription by binding to the gene promoter region [102]. MiRNAs can regulate the expression of ABC transporters and EMT markers. Interestingly, miR-200c and miR-145 have been shown to inhibit ABC transporters and suppress EMT [103], while miR-27a appears to upregulate ABC transporters and promote EMT [104].

MiRNAs can also assert their effect through other modalities, including the regulation of apoptosis and autophagy, control of anti-cancer drug metabolism, modulation of both drug targets and DNA repair, and regulation of GSH biosynthesis, as summarized by An et al. [105]. 

## 2. Emerging Strategies for Overcoming Resistance

### 2.1. Targeted Protein Degradation

Targeted protein degradation has emerged as a promising strategy to overcome drug resistance in various diseases, particularly cancer. This approach utilizes small molecules designed to selectively degrade disease-causing proteins. Among the key technologies in this area are proteolysis-targeting chimeras (PROTACs) and specific and nongenetic IAP-dependent protein erasers (SNIPERs).

#### 2.1.1. Proteolysis-Targeting Chimeras (PROTACs)

PROTACs are bifunctional molecules that recruit an E3 ubiquitin ligase to a target protein, marking it for degradation by the ubiquitin–proteasome system [106] (Figure 2). This method offers several advantages over traditional inhibition, such as the ability to target non-enzymatic proteins and those that have developed mutations, rendering them resistant to conventional inhibitors. For instance, PROTACs targeting the oncogenic fusion protein BCR-ABL1 have shown potential in overcoming resistance to kinase inhibitors in chronic myeloid leukemia (CML). By degrading BCR-ABL1, PROTACs can circumvent the resistance mechanisms that often emerge with kinase inhibitors, such as point mutations within the kinase domain [107,108]. PROTACs have also been shown to be effective in reversing resistance in prostate cancer to androgen receptor (AR) antagonists [109]. Similarly, BET (bromodomain and extra-terminal protein family)-targeted PROTACs have been found to potentially reverse resistance in castration-resistant prostate cancer and triple-negative breast cancer [110]. PROTACs targeting the EGF receptor (EGFR) for the treatment of non-small-cell lung carcinoma (NSCLC) have also shown strong promise in reversing resistance to EGFR inhibitors [111,112].

#### 2.1.2. Specific and Nongenetic IAP-Dependent Protein Erasers (SNIPERs)

SNIPERs utilize the cellular inhibitor of apoptosis proteins (IAPs) to facilitate the degradation of target proteins. Unlike PROTACs, SNIPERs do not necessarily require genetic modifications to the target cells, making them a versatile tool for targeting a range of disease-related proteins. The versatility of SNIPERs lies in their ability to induce degradation without relying on the ubiquitin–proteasome pathway, potentially overcoming resistance mechanisms that involve alterations in this pathway [113,114].

These approaches represent a paradigm shift in drug development, moving from inhibition to the complete removal of disease-causing proteins. The selective degradation of proteins offers the potential to address the challenges of drug resistance, particularly in cancers where resistance to targeted therapies is common.

### 2.2. Immunotherapy Combinations

Combining immunotherapy with other treatment modalities has shown significant promise in enhancing antitumor responses and overcoming resistance. Immune checkpoint inhibitors, such as pembrolizumab or nivolumab (anti-PD-1 antibodies), or atezolimumab (anti-PD-L1 antibody), have been significant additions in the advancement of cancer treatment by unleashing the immune system against tumors. While these immunotherapies have induced robust and lasting tumor responses, many patients will relapse over time due to the loss of T-cell function, lack of T-cell recognition, and development of escape mutation variants. As a result, treatments have transitioned to combination therapy.

#### 2.2.1. Combination with Chemotherapy

Combining immune checkpoint inhibitors with chemotherapy has demonstrated synergistic effects. Chemotherapy can induce immunogenic cell death, enhancing the release of tumor antigens and promoting a more robust immune response. In lung cancer, the combination of pembrolizumab with chemotherapy has led to improved survival outcomes compared to chemotherapy alone. Similarly, in bladder cancer, this combination has resulted in higher response rates and prolonged progression-free survival [115,116].

#### 2.2.2. Combination with Targeted Therapy

Targeted therapies, which inhibit specific molecular pathways critical for tumor growth, can also be combined with immune checkpoint inhibitors. This approach aims to target the cancer cells more precisely, while simultaneously enhancing the immune system’s ability to recognize and destroy these cells. For example, the combination of pembrolizumab with tyrosine kinase inhibitors (TKIs) has shown promising results in various cancers, including renal cell carcinoma and melanoma [117,118].

These combination strategies leverage the strengths of each modality, providing a multifaceted attack on tumors and addressing different mechanisms of resistance. By enhancing the overall antitumor response, these combinations hold the potential to overcome resistance that arises from monotherapies.

### 2.3. Precision Medicine

Precision medicine has been transformative in cancer treatment by tailoring therapies to the individual molecular profiles of patients’ tumors. Advances in genomic profiling and molecular diagnostics have enabled the identification of specific genetic alterations driving resistance, allowing for the development of personalized treatment strategies.

#### 2.3.1. EGFR T790M Mutation in NSCLC

A prime example of precision medicine in action is the management of non-small-cell lung cancer (NSCLC) patients with EGFR mutations. First-generation EGFR inhibitors, such as erlotinib and gefitinib, have been effective initially, but resistance often develops due to secondary mutations like EGFR T790M. The identification of this mutation has led to the development of third-generation TKIs like osimertinib, which specifically target the T790M mutation. Osimertinib has demonstrated significant efficacy in overcoming resistance, highlighting the impact of precision medicine on improving patient outcomes [119,120].

#### 2.3.2. Molecular Diagnostics

The integration of molecular diagnostics into clinical practice has enabled the identification of a wide range of actionable mutations and alterations. Techniques such as next-generation sequencing (NGS) provide comprehensive genomic profiles, guiding the selection of targeted therapies and combination treatments. This approach ensures that patients receive the most appropriate and effective therapies based on the unique characteristics of their tumors [121].

Precision medicine not only improves treatment efficacy but also minimizes unnecessary toxicity by avoiding treatments that are unlikely to benefit specific patients. By continuously evolving with advancements in genomic technologies, precision medicine remains at the forefront of overcoming resistance in cancer therapy.

### 2.4. Novel Drug Delivery Systems

Advances in drug delivery systems have opened new avenues for overcoming resistance by improving the pharmacokinetics and tumor-targeting capabilities of therapeutic agents. Nanotechnology-based systems, such as liposomes and nanoparticles, offer several benefits, including enhanced drug solubility, stability, and targeted delivery to tumor tissues.

#### 2.4.1. Liposomes

Liposomes are spherical vesicles with a phospholipid bilayer capable of encapsulating both hydrophilic and hydrophobic drugs. This encapsulation can protect drugs from degradation, reduce off-target effects, and improve drug biodistribution. For instance, liposomal formulations of chemotherapeutic agents like doxorubicin (e.g., Doxil) have shown improved efficacy and reduced cardiotoxicity compared to their free-drug counterparts. By enhancing drug accumulation in tumor tissues and reducing exposure to healthy tissues, liposomes can help overcome resistance mechanisms such as drug efflux pumps [122,123].

#### 2.4.2. Nanoparticles

Nanoparticles can be engineered to deliver drugs directly to tumor cells, exploiting the enhanced permeability and retention (EPR) effect commonly seen in tumors. These particles can be functionalized with ligands that target specific tumor-associated receptors, ensuring precise delivery of the therapeutic payload. Nanoparticles can also be designed to release drugs in response to specific stimuli within the tumor microenvironment, such as pH changes or enzymatic activity. This targeted delivery not only increases the concentration of the drug at the tumor site but also minimizes systemic toxicity, thereby addressing resistance mechanisms related to drug distribution and penetration [124,125].

These novel drug delivery systems represent a significant advancement in overcoming resistance, offering the potential to improve the efficacy and safety of cancer therapies. By enhancing drug delivery to tumor tissues and reducing off-target effects, these systems can effectively counteract various resistance mechanisms, ultimately improving patient outcomes.

### 2.5. Neurotherapy

Neurotherapy, or neuromodulation therapy, is increasingly being recognized for its potential role in overcoming resistance to cancer treatment. This approach involves the use of electrical, magnetic, or chemical means to modulate nervous system activity, which can have downstream effects on tumor biology and treatment efficacy. Techniques in neurotherapy include Deep Brain Stimulation (DBS), which is being explored for its potential to influence the tumor microenvironment and enhance immune system function; recent studies suggest that DBS can modulate immune responses and might be used in conjunction with immunotherapy to improve outcomes in cancer patients [126]. Transcranial Magnetic Stimulation (TMS) uses magnetic fields to stimulate nerve cells in the brain and is being studied for its ability to alter the brain’s microenvironment, potentially increasing the permeability of the blood–brain barrier and improving the delivery of chemotherapeutic agents to brain tumors [127]. Vagus Nerve Stimulation (VNS) has shown promise in modulating systemic immune responses; by stimulating the vagus nerve, researchers aim to enhance the body’s ability to mount an effective immune response against tumors [128]. These neurotherapeutic approaches are still in the experimental stages, but they offer exciting possibilities for enhancing the efficacy of existing cancer treatments and overcoming resistance.

### 2.6. Immuno-oncology

Immuno-oncology, or cancer immunotherapy, is yet another approach to cancer treatment that harnesses the body’s own immune system to target and destroy cancer cells. Despite significant successes, resistance to immunotherapy remains a challenge. Recent strategies to overcome this resistance include immune checkpoint inhibitors, such as pembrolizumab (anti-PD-1) and ipilimumab (anti-CTLA-4), which block proteins that inhibit immune responses. Combining these inhibitors with other treatments can enhance their effectiveness; for example, combining pembrolizumab with chemotherapy has shown improved survival rates in lung cancer patients [129]. Adoptive Cell Transfer (ACT) involves infusing patients with immune cells that have been engineered or expanded outside the body. CAR T-cell therapy, a type of ACT, has shown remarkable success in treating certain blood cancers by modifying T cells to better recognize and attack cancer cells [130]. Cancer vaccines aim to elicit an immune response against specific tumor antigens; recent advancements include personalized vaccines tailored to the unique mutations present in an individual’s tumor [131]. These approaches are continually evolving, with ongoing research focused on understanding and overcoming the mechanisms of resistance to immunotherapy.

More recently, mRNA vaccines have elicited significant interest in their potential for use in cancer treatment. mRNA vaccines consist of synthetic mRNA encoding tumor-associated or tumor-specific antigens. The antigen-specific mRNA is typically encapsulated in lipid nanoparticles or lipoplex formulated and delivered, in many cases, as injectable formulations. A number of these mRNA vaccines, including NCT03948763, NCT04573140, NCT04534205, among many others, are now in early-stage clinical development, being evaluated as stand-alone therapies or in combination with anti-PD1 or anti-PDL1 antibodies [132].

### 2.7. Biologics, Antibodies, and Cell Therapy

Biologics and cell therapies represent cutting-edge treatments that offer targeted approaches to overcoming drug resistance in cancer. Monoclonal antibodies are engineered to target specific proteins in cancer cells; for instance, trastuzumab targets the HER2 protein in breast cancer, and recent biosimilars have been developed to improve accessibility and affordability [133]. Bispecific antibodies are designed to engage two different targets simultaneously; for example, blinatumomab brings T cells into close proximity with cancer cells, enhancing the immune response [134]. CAR T-cell therapy involves modifying a patient’s T cells to express chimeric antigen receptors that specifically target cancer cells; it has been highly effective in treating certain types of leukemia and lymphoma. Recent advancements aim to enhance the persistence and efficacy of CAR T cells, as well as expand their use in solid tumors [135]. Tumor-infiltrating lymphocyte (TIL) therapy involves harvesting immune cells from a patient’s tumor, expanding them in the lab, and re-infusing them to attack the cancer; this approach has shown promise in treating melanoma and other cancers [135,136]. These biologics and cell therapies are at the forefront of personalized cancer treatment, offering new ways to overcome resistance and improve patient outcomes.

## 3. Pharmaceutical Approaches to Cancer Drug Resistance and Combination Immunotherapy

In the landscape of cancer treatment, big pharmaceutical companies have implemented several commercial practices aimed at addressing the pervasive issue of drug resistance. These strategies are intricately linked to the development and marketing of innovative therapies designed to overcome the various mechanisms that cancer cells employ to resist treatment. One prominent approach is the promotion and advancement of targeted protein degradation technologies, such as proteolysis-targeting chimeras (PROTACs) and specific and nongenetic IAP-dependent protein erasers (SNIPERs) [113]. These technologies represent a paradigm shift from traditional small-molecule inhibitors by facilitating the degradation of disease-causing proteins that have developed resistance mechanisms. For instance, PROTACs have shown efficacy in degrading oncogenic fusion proteins like BCR-ABL in chronic myeloid leukemia, thereby circumventing resistance mechanisms often encountered with kinase inhibitors [137]. By investing in and promoting these technologies, pharmaceutical companies aim to expand their therapeutic portfolios and improve patient outcomes through novel mechanisms of action.

Furthermore, pharmaceutical companies are actively engaging in the development and commercialization of combination therapies, particularly in the realm of immunotherapy. Immunotherapy, such as immune checkpoint inhibitors like pembrolizumab, has revolutionized cancer treatment by harnessing the body’s immune system to target tumors. Companies are strategically combining immunotherapy with conventional chemotherapy or targeted therapies to enhance treatment efficacy and overcome resistance mechanisms [138]. This approach not only broadens the therapeutic spectrum but also capitalizes on synergistic effects between different modalities. By leading clinical trials and investing in robust research pipelines, these companies seek to establish new standards of care that address the complex and evolving nature of cancer resistance, thereby solidifying their positions as leaders in oncological innovation and therapy development.

### Clinical Implications and Future Directions

Understanding cancer resistance mechanisms and advancing strategies to overcome this challenge are pivotal for improving cancer therapy outcomes. Personalized treatment approaches, informed by genomic profiling and molecular diagnostics, play a crucial role in targeting specific resistance mechanisms such as drug efflux, target alterations, and DNA repair pathways. This tailored approach ensures the selection of therapies that are likely to be effective for individual patients, thereby enhancing treatment success rates [117,139,140].

Combination therapies represent a promising strategy to synergistically enhance antitumor responses and mitigate resistance as shown in Table 1. For example, combining immune checkpoint inhibitors with chemotherapy has shown efficacy in improving survival outcomes in lung and bladder cancers [141,142]. Similarly, combining pembrolizumab with tyrosine kinase inhibitors has demonstrated promising results in renal cell carcinoma and melanoma [143,144]. These combinatorial approaches offer multifaceted strategies to combat different resistance mechanisms and improve overall treatment responses.

Of the various modalities in combating drug resistance, combination therapies have emerged as a primary approach to overcome resistance. By using a synergistic blend of drugs, these therapies aim to target multiple pathways simultaneously, thus overcoming resistance and improving patient outcomes. Several notable combinations are currently in late-stage clinical trials, highlighting their potential in treating different types of cancer. Table 2, below, provides a summary of some of these combination therapies, their target cancer types, and clinical trial stage.

In addition to combination therapy, other novel therapeutic approaches, such as targeted protein degradation, hold potential in revolutionizing cancer treatment paradigms. Technologies like PROTACs and SNIPERs enable selective degradation of disease-causing proteins, overcoming resistance mechanisms that conventional inhibitors may not address effectively [145,146]. These advancements offer a new frontier in cancer therapy by targeting proteins that have developed resistance through mutations or alterations.

Advanced drug delivery systems also present significant clinical implications in overcoming drug resistance. Systems like liposomes and nanoparticles enhance the delivery and bioavailability of therapeutic agents, improving their accumulation in tumor tissues while reducing systemic toxicity. By addressing pharmacokinetic challenges and enhancing drug-targeting capabilities, these delivery systems can improve treatment efficacy and overcome resistance mechanisms [147,148].

Future directions in cancer therapy should focus on optimizing combination strategies, developing next-generation therapeutics, and enhancing drug delivery systems. Continued advancements in genomic technologies and molecular diagnostics will refine precision medicine approaches, identifying a broader range of resistance mechanisms and actionable targets [156,157]. Innovations in next-generation therapeutics, including fourth-generation inhibitors and novel protein degraders, will be crucial for staying ahead of evolving resistance mechanisms [158,159].

The translation of these emerging strategies from bench to bedside requires collaborative efforts between researchers, clinicians, and pharmaceutical industries. Emphasizing translational research and conducting robust clinical trials will expedite the development and implementation of these innovative therapies in clinical practice [160,161]. By addressing the complexities of cancer resistance comprehensively, the field of oncology holds promise for delivering more effective and personalized treatments, ultimately improving outcomes for patients with resistant malignancies.

## 4. Future Outlook

Future cancer research should focus on understanding and overcoming therapy resistance by investigating the molecular mechanisms of drug efflux, target alterations, DNA repair, cell death inhibition, and tumor microenvironment changes. Advanced genomic, proteomic, and metabolomic technologies are crucial for identifying novel resistance pathways and targets [162,163]. Emphasizing the development of combination therapies that address multiple resistance mechanisms concurrently, such as integrating chemotherapies with targeted therapies, immunotherapies, or epigenetic modifiers, could enhance treatment efficacy [164]. Regarding strategies to combat cancer resistance, the synthesis of new metal complexes (e.g., Cu, Ru, Rh, Au, Pt, etc.) can play an important role in overcoming certain limitations of established therapies [165]. These metal complexes have shown promise in enhancing the efficacy of existing treatments and mitigating resistance by targeting multiple pathways simultaneously. Their unique chemical properties allow them to interact with biological molecules in novel ways, potentially leading to more effective and less toxic cancer therapies. Therefore, the exploration and development of these metal complexes should be considered as potentially important in cancer treatment. Additionally, personalized medicine approaches using patient-specific pharmacogenomics data will enable more precise and effective treatment strategies, potentially reducing resistance and improving patient outcomes [166,167]. The integration of artificial intelligence (AI) into cancer research holds significant potential for analyzing large datasets, identifying patterns, and predicting resistance mechanisms, thereby accelerating the discovery of effective treatments [168,169].

## 5. Discussion

In our review, we explored the multifaceted mechanisms underlying cancer resistance, highlighting key areas, such as drug efflux, target alterations, DNA repair, cell death inhibition, evasion of drug action, microenvironmental changes, and epithelial-to-mesenchymal transition. Each of these mechanisms contributes to the complexity of treatment resistance, posing significant challenges to current therapeutic strategies. Understanding these resistance pathways is crucial for developing more effective treatments. While targeted therapies and immunotherapies have shown promise, their efficacy is often limited by the tumor’s adaptive mechanisms. This underscores the need for continued research into the molecular underpinnings of resistance and the development of novel therapeutic approaches that can circumvent or counteract these mechanisms.

## 6. Conclusions

In conclusion, overcoming cancer resistance requires a comprehensive understanding of the underlying mechanisms and the development of innovative treatment strategies. Future research should prioritize the elucidation of the molecular basis of drug resistance and leverage advanced technologies to discover new targets. The integration of artificial intelligence (AI) into cancer research holds significant potential for analyzing large datasets, identifying patterns, and predicting resistance mechanisms, thereby accelerating the discovery of effective treatments. Developing combination therapies that address multiple resistance pathways and employing personalized medicine approaches will be critical in improving treatment outcomes. By focusing on these areas and incorporating AI-driven insights, we can make significant strides in the fight against cancer, enhance the efficacy of therapeutic interventions, and overcome drug resistance.

## Figures and Tables

**Figure 1 biomedicines-12-01801-f001:**
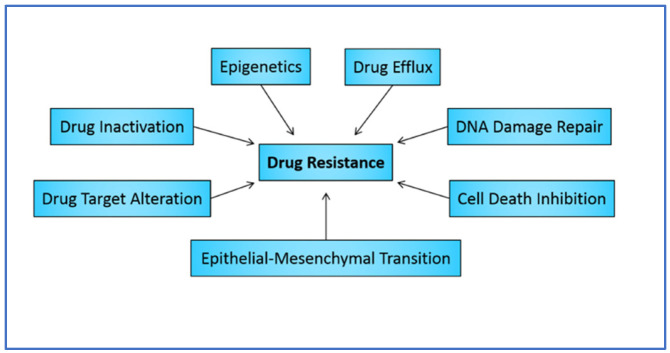
Different modes of drug resistance.

**Figure 2 biomedicines-12-01801-f002:**
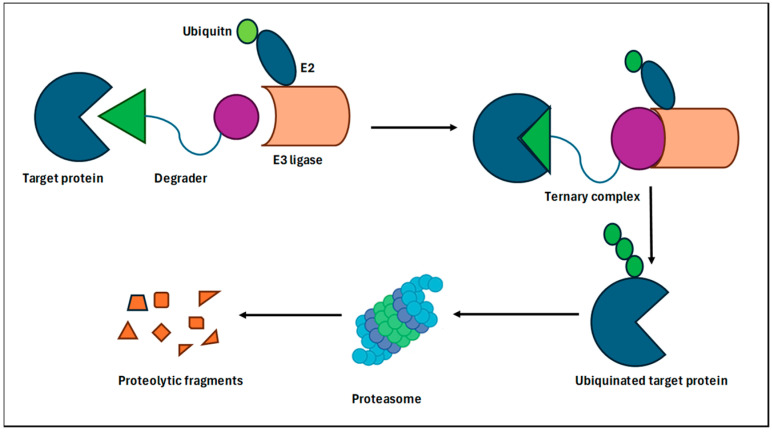
Mechanism of PROTAC and protein degradation.

**Table 1 biomedicines-12-01801-t001:** Clinical outcomes of combining therapies.

Combination Therapy	Cancer Type	Clinical Outcome	Reference Numbers
Pembrolizumab + Chemotherapy	Lung cancer	Improved survival outcomes	[141]
Pembrolizumab + Chemotherapy	Bladder cancer	Higher response rates, prolonged progression-free survival	[141]
Pembrolizumab + TKIs	Renal cell carcinoma	Promising results in clinical trials	[143,144]
Pembrolizumab + TKIs	Melanoma	Improved overall survival and response rates	[143,144]
Trastuzumab + Pertuzumab	HER2-positive breast cancer	Enhanced progression-free survival	[145,146]
Ipilimumab + Nivolumab	Melanoma	Increased objective response rate	[147,148]

**Table 2 biomedicines-12-01801-t002:** Clinical trials of combination therapies.

Drug	Combination Therapy	Cancer Type	Clinical Stage	Reference
Pembrolizumab	Chemotherapy (Carboplatin and Pemetrexed)	Non-Small Cell Lung Cancer (NSCLC)	Phase 3	[149]
Pembrolizumab	Lenvatinib	Renal Cell Carcinoma	Phase 3	[150]
Trastuzumab	Pertuzumab, Docetaxel	HER2-Positive Breast Cancer	Phase 3	[151]
Nivolumab	Ipilimumab	Melanoma	Phase 3	[152]
Atezolizumab	Bevacizumab, Chemotherapy (Carboplatin and Paclitaxel)	Hepatocellular Carcinoma	Phase 3	[153]
Durvalumab	Tremelimumab, Chemotherapy (Cisplatin and Etoposide)	Non-Small Cell Lung Cancer (NSCLC)	Phase 3	[154]
Ibrutinib	Venetoclax	Chronic Lymphocytic Leukemia	Phase 3	[155]

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
