# Peer review of "Overcoming Cancer Resistance: Strategies and Modalities for Effective Treatment"

_biomedicines, 2024, doi:10.3390/biomedicines12081801_

Round 1

Reviewer 1 Report

Comments and Suggestions for Authors

The review article by DiPaola and Koirala describes the possible ways to overcome the challenges faced in cancer treatment due to drug resistance. In my opinion its a nice piece of work and would be worth to appear in Biomedicines after a minor revision.

1. I would suggest that authors should include a comprehensive Future Outlook specifically pointing out that what should be the focus of future research work and how it could be achieved? In that way some of the possible strategies that have been discussed in Discussion and Conclusion part could be moved out. 

2. Line 81, please remove one "such" as it appears twice in the same sentence.

3. Line 210, please omit double space between, "microenvironment" and "can".

4. Please pay attention to formatting issues. For instance, each work in some of the subtitles begin with capital letters but in others not. For instance, subtitles under section 2.1 and 2.2.

Reviewer 2 Report

Comments and Suggestions for Authors

The manuscript is a review about a very current and interesting topic: cancer resistance. The authors approach  both mechanisms of cancer resistance and strategies for overcoming resistance. I believe that the article is of great interest to the readers of this journal and I recommend that it be published after a minor revision.

As comments/suggestions:

1. The introduction should be more elaborate and the period of the articles considered in writing this review should also be specified.

2. Regarding strategies to combat cancer resistance, the synthesis of new metal complexes (e.g. Cu, Ru, Rh, Au, Pt, etc.) has an important role to overcome the problems of those established in therapy. So they should also be mentioned in the review

3. If among the mentioned strategies there are some that are already in clinical trials, they must be mentioned

4. In  my opinion, the number of references from the last 5-7 years should predominate since it is a review
